# Investigating healthcare worker mobility and patient contacts within a UK hospital during the COVID-19 pandemic

Jared K. Wilson-Aggarwal [1], Nick Gotts[1], Wai Keong Wong [2], Chris Liddington[2], Simon Knight[2], Moira J. Spyer[3,4], Catherine F. Houlihan[3,4], Eleni Nastouli [3,4] & Ed Manley [1✉]

### Abstract

**Background** Insights into behaviours relevant to the transmission of infections are extremely valuable for epidemiological investigations. Healthcare worker (HCW) mobility and patient contacts within the hospital can contribute to nosocomial outbreaks, yet data on these behaviours are often limited.

**Methods** Using electronic medical records and door access logs from a London teaching hospital during the COVID-19 pandemic, we derive indicators for HCW mobility and patient contacts at an aggregate level. We assess the spatial-temporal variations in HCW behaviour and, to demonstrate the utility of these behavioural markers, investigate changes in the indirect connectivity of patients (resulting from shared contacts with HCWs) and spatial connectivity of floors (owing to the movements of HCWs).

**Results** Fluctuations in HCW mobility and patient contacts were identified during the pandemic, with the most prominent changes in behaviour on floors handling the majority of COVID-19 patients. The connectivity between floors was disrupted by the pandemic and, while this stabilised after the first wave, the interconnectivity of COVID-19 and non-COVID-19 wards always featured. Daily rates of indirect contact between patients provided evidence for reactive staff cohorting in response to the number of COVID-19 patients in the hospital.

**Conclusions** Routinely collected electronic records in the healthcare environment provide a means to rapidly assess and investigate behaviour change in the HCW population, and can support evidence based infection prevention and control activities. Integrating frameworks like ours into routine practice will empower decision makers and improve pandemic preparedness by providing tools to help curtail nosocomial outbreaks of communicable diseases.

### Plain language summary

Movement of healthcare workers and their patient contacts can contribute to outbreaks of infection in the healthcare environment. We use electronic medical records and door access logs from a London hospital to derive indicators for staff behaviour during the COVID-19 pandemic. Changes in staff behaviour were most prominent on floors handling the majority of COVID-19 patients. We also show how the flow of staff between COVID-19 and non-COVID-19 wards continued throughout the pandemic, but find evidence that indirect contact between COVID-19 positive and negative patients reduced as COVID-19 prevalence increased. We suggest these routinely collected data on HCW behaviour should be used to support decision makers in activities to help curtail disease outbreaks in healthcare settings.

[1] School of Geography, University of Leeds, Woodhouse, Leeds LS2 9JT, UK. [2] Department of Digital Healthcare Technology, University College London Hospitals NHS Foundation Trust, London, UK. [3] Department of Infection, Immunity and Inflammation, UCL GOS Institute of Child Health University College London, London, UK. [4] Department of Clinical Virology, University College London Hospitals NHS Foundation Trust, London, UK. ✉email: E.J.Manley@leeds.ac.uk

Human mobility and contact are significant drivers for the transmission of communicable diseases, such as severe acute respiratory syndrome coronavirus 2 (SARS-CoV-2) that resulted in the COVID-19 pandemic[1]. While passively collected mobile phone and app-derived GPS trajectory data provide an indication of populations' mobility and social mixing patterns[2], only broad regional generalisations can be drawn. Transmission of SARS-CoV-2 occurs through close proximity between infectious and susceptible individuals due to either direct contact or respiratory aerosols in the shared space[3]. Therefore, insights into behaviours at a fine scale, such as within indoor environments, are also required to deepen our understanding of behaviours associated with the transmission of infections, and improve our ability to identify and prevent transmission events. This is particularly relevant for healthcare settings, where infection outbreaks present a significant risk to vulnerable patients through increased morbidity and mortality.

The concern in relation to infection transmission within hospital environments extends more widely than COVID-19, and includes other healthcare-associated infections (HAIs). The impact of HAIs on healthcare systems is considerable, resulting in staff illness, complications in patient outcomes and increasing healthcare costs. In England between 2016 and 2017, HAIs were estimated to have caused > 28,000 deaths, contributed to 21% of hospital bed days, resulted in > 79,000 days of absence in frontline HCWs and cost the NHS an estimated £2.7 billion[4]. The surveillance prevention and control of HAIs is a challenge as granular data are often limited and the transmission pathways are highly variable; dependent on the epidemiology of the pathogen, be it bacterium, virus or fungus[5,6].

Nosocomial infections in patients are well defined, as are frameworks for their prevention and control[7]. To manage HAIs in patients, practitioners responsible for infection prevention and control (IPC) frequently use passive data sources that are routinely collected, such as medical records. These data sources provide information on the patient's location within the hospital and their contacts with staff, which can be used to support surveillance, mapping patient trajectories and contact tracing[8–10]. Historically these data sources have been handled manually, using time intensive frameworks that prevent their use in real-time. Hospitals that have moved to digital systems have seen an increase in the effectiveness and efficiency of patient-focused IPC, through improved availability of data resources and reduced burdens of manual data collection and processing[11,12]. However, while these data streams are well established for patient-focused activities, those for the management of staff infections are relatively underdeveloped. This is surprising given that, like patients, HCWs are at risk of both acquiring and facilitating the transmission of HAIs[13].

The rapid spread of SARS-CoV-2 has emphasized the need to protect front-line HCWs. Early in the pandemic the prevalence of COVID-19 infection for HCWs was high, with one London hospital reporting infection in 44% of HCWs[14] and a global estimation of 11% of HCWs infected with the virus[15]. What's more, the risk of infection for HCWs varied between roles and spatially, with a higher risk of infection for those working in non-emergency wards and for nurses[15]. Nosocomial outbreaks of SARS-CoV-2 result from a small number of highly infectious individuals, and transmission chains may include HCWs among the likely super-spreaders[16]. Behavioural processes, such as contact and mobility patterns, generate heterogeneity in the transmission of communicable diseases[1,17] and, similar to the management of HAIs in patients, passively collected data on the within-hospital behaviours of HCWs can contribute to a more informed and rapid response to outbreaks.

HCW behaviours have been investigated using surveys[18], observations[19,20] and tracking technologies[21–24], but these data collection methods are often prohibitively time intensive, expensive, or only provide a snapshot view that is not hospital wide. Electronic medical records (EMRs) have also been previously used to investigate HCW space use and patient contacts[25–27], but they are either optimised for reconstructing patient trajectories or suffer from high spatial uncertainty. Additional databases, such as door access logs could complement EMRs by supplementing spatiotemporal information on HCW mobility. These data sources are analogous in nature to the passively-collected spatial data from mobile phone records, which were used during the COVID-19 pandemic to demonstrate the effectiveness of movement restrictions in reducing contact rates, and subsequently lowering levels of community transmission[28]. Using the routinely collected hospital data as an indicator for HCW behaviour provides opportunities to enhance evidence-based IPC in a similar way; supporting contact tracing efforts, validating transmission pathways and helping to monitor the effectiveness of interventions in the hospital.

As with other communicable diseases, IPC interventions to prevent nosocomial outbreaks of COVID-19 include hand washing, the use of personal protective equipment (PPE), limiting the traffic of people in the hospital, and cohorting staff and patients[29]. The routinely collected data cannot identify or monitor all HCW behaviours that are epidemiologically relevant, but can indicate their level of space use within the hospital, their frequency of movement, the number of patients they contact and the frequency of patient contact. As behavioural markers these metrics provide quantitative measures for IPC interventions aimed specifically at reducing the spatial connectivity of spaces (e.g. by restricting staff access/flow to areas) and social connectivity of individuals in the hospital (e.g., through patient and staff cohorting). The data can therefore be used to assess the extent to which interventions targeted towards HCW mobility and patient contacts have been successful in achieving their aim, or in determining opportunities for improvement.

This paper outlines a framework for the use of routinely-collected hospital data in the measurement of HCW behaviour at an aggregate level. We describe (1) the integration of diverse digital data sources for the quantification of HCW mobility and patient contact within the hospital setting, and (2) demonstrate the use of these data sources in supporting IPC activities through a series of analyses. Specifically, we use data from a London Hospital to investigate whether or not (i) HCW mobility, (ii) HCW patient contacts, (iii) spatial connectivity (flow between floors) and (iv) indirect contacts between patients (through shared HCW contacts) were reactive to the early stages of the COVID-19 pandemic. We find that fluctuations in HCW mobility and patient contacts were most prominent on floors handling the majority of COVID-19 patients, while the flow of HCWs between COVID-19 and non COVID-19 wards continued throughout the pandemic, and daily rates of indirect contact between patients provided evidence for reactive staff cohorting.

## Methodology

**Study site and context**. University College London Hospital NHS Trust (UCLH) is a large acute and tertiary referral academic hospital located in central London. The Main UCLH building is comprised of a central Tower that has 19 floors (floors −2 to 16) and is linked to two other buildings; the Podium and the Elizabeth Garett Anderson (EGA) Wing. In this analysis, we only considered data for the Tower building at UCLH. Here we describe floors within the Tower by the ward/department that predominantly occupies it; the basement (floor-2), imaging (floor -1), emergency department (ED on floor 0), acute medicine unit (AMU on floor 1), day surgery (floor 2), critical care (floor

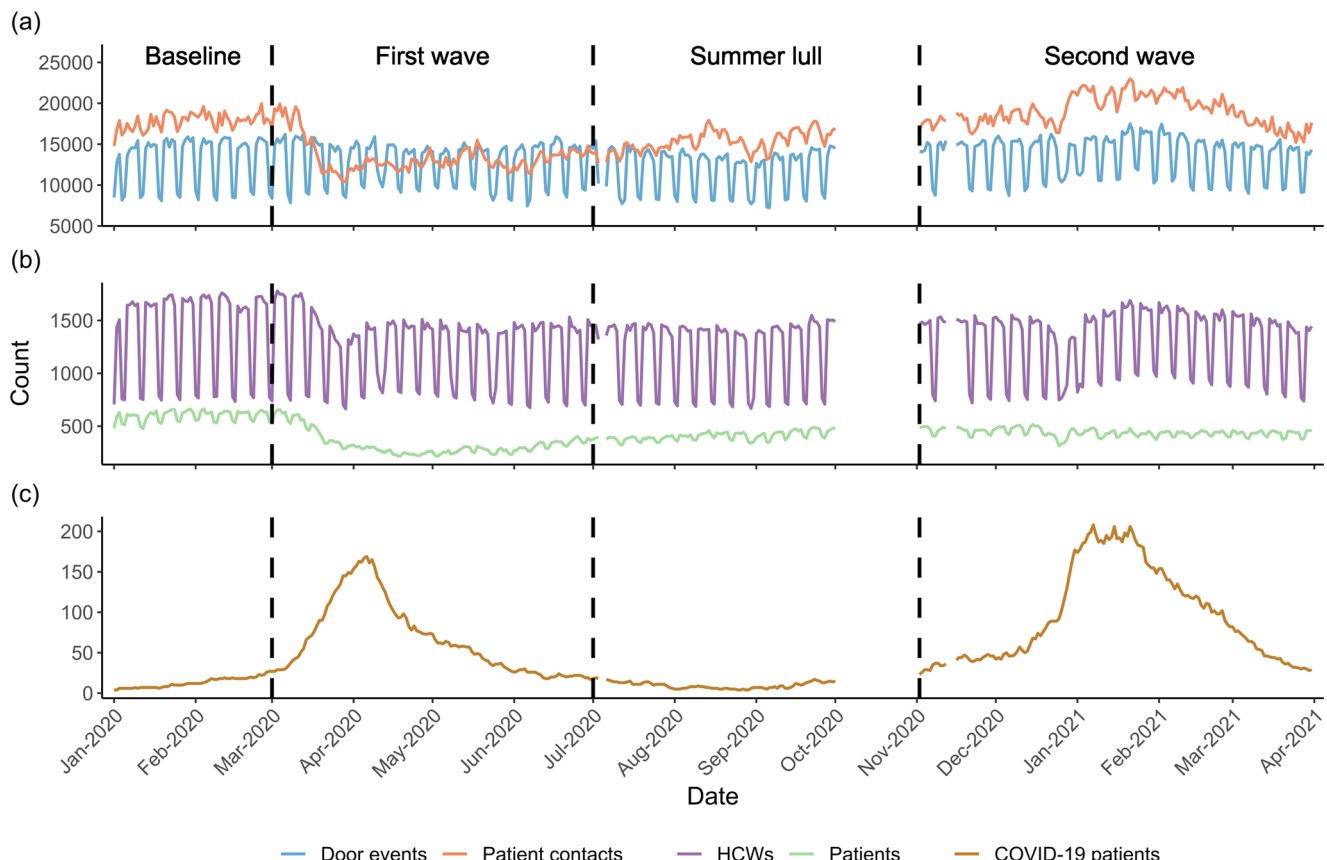

**Fig. 1 Daily counts for indicators of healthcare worker behaviour, staffing levels and patient numbers in the Tower building of University College London hospital during the first year of the COVID-19 pandemic.** Plot (**a**) shows the activity of healthcare workers in the hospital as characterised by the daily count of door events (blue line) and patient contacts (red line) logged in routinely collected electronic data sources. Plot (**b**) shows daily counts for the number of healthcare workers (purple line) and patients (green line) identified in the data. Plot (**c**) shows the number of COVID-19 patients in the hospital (gold line), which was used to determine the different stages of the pandemic. For each plot the vertical black dashed lines indicate the start and stop points for each stage of the pandemic; Baseline (pre-pandemic), first wave, summer lull and second wave. Data for October 2020 was not available.

3), plant (floor 4), nuclear medicine (floor 5), short stay surgery (floors 6), hyper-acute stroke unit (HASU on floor 7), respiratory & infectious diseases (floor 8), general surgery (floor 9), care of the elderly (CoE on floor 10), paediatrics (floor 11), adolescents (floor 12), oncology (floor 13), head and neck (floor 14), private wards (floor 15) and haematology (floor 16).

During the pandemic the UCLH Tower became a key site for COVID-19 care in London, and the peaks in the number of COVID-19 patients in the Tower (Fig. 1c) closely followed that reported for all London hospitals (as downloaded from gov.uk; $r = 0.97$). To investigate changes in the daily number of events in different stages of the pandemic, we manually identified distinct time periods based on the number of COVID-19 patients in the Tower. The first stage of the pandemic was March 1st– June 30th 2020, when the 'first wave' of COVID-19 hospital admissions was experienced, during which the WHO declared a pandemic (March 11[th] 2020) and the first national lockdown in England was announced (23[rd] March 2020). The second stage was between July 1st and August 31st 2020, which represented the 'summer lull', where the number of COVID-19 patients in London hospitals remained at a low level and community interventions were eased. The third stage was between November 1st 2020 and March 31st 2021, when the 'second wave' of COVID-19 hospital admissions occurred, the second national lockdown was announced (5[th] November 2020) and the mass-vaccination programme began (December 8[th] 2020).

We also identified a pre-pandemic or 'baseline' period between January 1st and February 28th 2020. This time period was prior to

the substantial rise in COVID-19 admissions in the hospital, and was considered to be when 'normal' working patterns (HCW movement and patient contacts) would be observed. We use this baseline period to compare the behaviour of HCWs during the pandemic to that pre-pandemic. Using data from corresponding weeks of the year may provide a more appropriate baseline when conducting such an analysis outside of the pandemic; as seasonal patterns in hospital admissions that likely influence HCW behaviour are known to occur[30]. However, the pandemic was associated with community interventions that disrupted seasonal patterns in hospital admissions concerning non-COVID-19 diseases[31,32]. Therefore, our baseline period provides an appropriate representation of 'normal' working patterns given the non-COVID-19 patient population was not comparable to that pre-pandemic, and so typical seasonal variations in staff behaviour were not expected.

**Data sources.** The data sources used in this study were selected on the basis of providing spatial and temporal indicators of staff movement and patient contacts within the hospital. For clarity, Table 1 provides definitions of terms relating to behavioural processes that are investigated in this study. The study protocol was approved by the NHS Health Research Authority (South Central – Berkshire REC ref: 20/SC/0147; protocol number: 130861) and ethical oversight was provided by the UCLH research ethics committee (IRAS project ID: ref. 281836; UCL

**Table 1 Definitions and data sources for behavioural processes relating to healthcare worker activity in the healthcare environment.**

| Behavioural process | Definition | Data source |
|---|---|---|
| Mobility | The frequency of movement exhibited by individuals/populations between discrete locations. | Security door access logs |
| Direct patient contact | Face to face interactions between healthcare workers and patients. | Electronic medical records |
| Indirect contact | The secondary contact between individuals resulting from their direct contact with others. | Electronic medical records |
| Social connectivity | The relationship between individuals, as determined by their direct and/or indirect contacts. | Electronic medical records |
| Spatial connectivity | The relationship between discrete locations, as determined by the spatial activity of individuals. | Security door access logs & electronic medical records |

GOS ICH R&D approval number: 20PL06). Data were extracted and deidentified by data managers at UCLH, and informed consent was waivered.

Patient contact events in EMRs were extracted from Epic, a privately owned hospital system used by UCLH for managing medical records. While Epic contains a large volume of data on patient diagnosis and treatment, we only use specific fields that provide information on the spatial and temporal attributes of within-hospital contacts between staff and patients. Data fields included the datetime of events, a description of the location (bed ID and floor), an indicator of the event type, anonymous identifiers for the patient, pseudonymous identifiers for the HCW and the COVID-19 status of the patient at the time of the event (0/1).

Door events were extracted from the database for security door access logs, known in this context as CCure. The security doors were located at entry/exit points of wards and access cards were also used to operate staff lifts. Data fields included the datetime of events, a description of the location (door ID and floor), direction of passage (in/out), status (accepted/rejected) and a pseudonymous staff identifier.

**Data cleaning**. All data processing was conducted in R[33]. Data for events outside the UCLH Tower were discarded. Data for the month of October 2020, a weekend in July and another in November were also discarded, as records either could not be extracted or had an unusually low number of events (indicating an issue with extraction). Contact events in EMRs that did not require face to face contacts (e.g., Telephone or Letter) were excluded. Door events with a rejected status were removed along with duplicate events in the same direction that were within 60 seconds of each other. Two types of lift (or elevator) events were present in the door access logs; Lift Calls where a card is used to request a lift, and Lift commands where a card is used before selecting which floor to go to. All Lift Call events were removed as they overinflate the number of movement events for individuals using lifts (because some individuals may make multiple and repeated lift calls while waiting for a lift).

**Aggregate measures**. Staffing levels, $|H|$, were determined by summing the number of distinct HCWs, $h$, in the set of HCW IDs identifiable in the routinely collected data, $H$. Staffing levels were calculated for each day, $t$, stage of the pandemic, $Stage$, and for each floor, $f$, and the entire building. For each stage of the pandemic the mean daily staffing levels, $\bar{H}$, were calculated for the entire building and each floor. To account for the weekly pattern in staffing levels (see Fig. 1 of the results), we calculated the means separately for weekdays and weekends.

Door events, $m$, were used as an indicator of HCW behaviour in terms of their mobility, where $M$ is the full set of door events.

The number of door events, $|M|$, was used as an absolute measure of HCW mobility and was calculated for the entire building and each floor on each day. There was a strong correlation between daily HCW mobility and daily staffing levels ($r = 0.94$) and therefore, to control for changes in staffing levels, the rate of mobility, $Mr$, was calculated as a function of staffing levels, where:

$$Mr_{t,f} = \frac{|M_{t,f}|}{|H_{t,f}|} \tag{1}$$

To compare mobility levels between stages of the pandemic, the mean daily mobility, $\bar{M}$, and the mean daily rate of mobility, $\overline{Mr}$, were calculated for the entire building and for each floor during the different stages of the pandemic. Again, due to the weekly temporal pattern, these means were calculated separately for weekdays and weekends.

Patient levels, $|P|$, were calculated by summing the number of distinct patients, $p$, in the set of patient IDs identifiable in the data, $P$, for each day, and the mean daily patient levels, $\bar{P}$, calculated for the entire building, and for each floor, during each stage of the pandemic. The same metrics were also calculated using the subset of patients known to be positive for COVID-19, $P^{Positive}$.

Patient contact events, $c$, were used as an indicator of HCW behaviour in terms of patient engagement where $C$ is the full set of patient contacts. The number of patient contacts, $|C|$, was used as an absolute measure and was calculated for the entire building and each floor on each day. There was a correlation between daily patient contacts and daily patient levels ($r = 0.65$), and therefore we also calculated the daily rate of patient contacts, $Cr$, as a function of patient levels, where:

$$Cr_{t,f} = \frac{|C_{t,f}|}{|P_{t,f}|} \tag{2}$$

To compare levels of patient engagement between stages of the pandemic, the mean daily number of patient contacts, $\bar{C}$, and the mean daily rate of patient contacts, $\bar{Cr}$, were calculated for the entire building and for each floor during the different stages of the pandemic.

To investigate the weekly and hourly patterns of mobility and patient engagement, a count for the number of door events and patient contacts was made for each hour, $hr$, of each day of the week, $w$, and separately for the different stages of the pandemic. These counts were then weighted by dividing them by the number of days each day of the week appeared in the dataset.

**Changes in time and space**. To investigate how the measures of daily HCW behaviour, staffing levels and patient levels differed from that pre-pandemic (baseline), on each floor and within the entire building, the normalised difference to baseline, $N$, was calculated for each day e.g. normalised difference to baseline for

HCW mobility across the entire building, $N_t^M$, where:

$$N_t^M = \frac{(|M_t| - \bar{M}_{Baseline})}{\bar{M}_{Baseline}} \qquad (3)$$

The normalised difference to baseline was also calculated for the averaged values for HCW behaviour during each stage of the pandemic. $N$ can be interpreted as proportional change, but is presented as percentage change in the results. For metrics with a strong weekly pattern ($H$, $Mr$ and $M$), $N$ for weekdays was calculated using the weekday average, and the weekend average was used for weekends.

**Spatial connectivity.** A dyadic analysis was conducted for each time period to assess the relationship between floors in terms of the flow of HCWs between them. For each spatial dyad (e.g. floors 1 & 2, floors 1 & 3 etc.) and using both door events and patient contacts, the number of HCWs that were active on both floors in any single day was extracted where:

$$dyad_{t,f,i} = |H_{t,f} \cap H_{t,i}| \qquad (4)$$

The index of the second floor in the dyad is denoted $i$. The resulting matrix was then treated as a weighted network with the diagonal set to zero. Lift events were excluded from this analysis as it was not possible to identify the floor on which they occurred. The Louvain clustering algorithm was used to identify floors that had stronger links. Louvain clustering uses a deterministic algorithm with a hierarchical greedy modularity maximization-based approach[34]. To check the robustness of clusters we also identified clusters using the leading eigenvalue and walktrap algorithms. All clustering analyses were implemented using the R package igraph v1.2.7[35].

**Patient connectivity.** Patient connectivity, $S$, was determined by identifying the number of COVID-19 negative patients each patient was indirectly in contact with through shared contacts with the same HCWs on the same day. This was achieved by first identifying the set of HCWs that had contact with the $j$th patient on each day, where:

$$H_{j,t} = \{h : C_t(p_j, h) = 1\} \qquad (5)$$

Next the set of patients not known to be positive for COVID-19, $P^{Negative}$, and that had also been in contact with any of the HCWs in $H_{j,t}$ were identified, where:

$$P_{j,t}^{Negative} = \left\{ p : p \in P_t^{Negative}, p \neq p_j, \exists h \in H_{j,t} | C_t(p, h) = 1 \right\} \qquad (6)$$

$S$ was then calculated for each patient as a proportion of all patients not known to be positive for COVID-19, and expressed as a percentage in the results where:

$$S_{j,t} = \frac{|P_{j,t}^{Negative}|}{|P_t^{Negative}|} \qquad (7)$$

For each day and stage of the pandemic, we made separate calculations for the average patient connectivity of patients not known to be positive for COVID-19, $\bar{S}^{Negative}$, and COVID-19 positive patients, $\bar{S}^{Positive}$.

**Statistics and reproducibility.** Linear models were used to statistically determine if daily metrics for staffing levels, patient levels, HCW mobility and patient contacts during each stage of the pandemic were different to baseline. Separate models were run for each metric, with the daily values as the response variable and the stage of the pandemic as the only fixed effect.

To investigate whether or not the $N$ for daily rates of HCW mobility and patient contacts were different on COVID-19 floors compared to non-COVID-19 floors, a mixed effects linear model was run with $N$ as the response variable. An interaction term was included between the stage of the pandemic and a binomial flag for whether the floor handled the majority of COVID-19 patients. Floor ID was included as a random effect. Data during the baseline were not included in the model.

Linear models were used to identify differences in $N_{t,f}^{Mr}$ and $N_{t,f}^{Cr}$ compared to those at baseline, and these had an interaction term between the stage of the pandemic and floor ID. To investigate the relationship between the daily number of COVID-19 patients in hospital and $N$ (for daily rates of HCW mobility and patient contacts), linear models were run with $N$ as the response variable and an interaction term between the stage of the pandemic, floor ID and the logged (base 2) daily number of COVID-19 patients. For COVID-19 floors, the number of COVID-19 patients was taken as the number of patients on the floor, while for non-COVID-19 floors the number of COVID-19 patients in the entire hospital was used.

To assess the relationship between the daily connectivity of patients and the number of COVID-19 patients in the hospital during each stage of the pandemic, we used a linear model with an interaction term between the logged (base 2) number of COVID-19 patients in the hospital and the stage of the pandemic.

The emmeans package (v1.3.3) was used to extract specific post-hoc comparisons of interest (e.g. baseline vs first wave and baseline vs summer lull), and the package lme4 (v1.1-25) was used to run mixed effects models. The DHARMa package (v0.4.1) was used for model diagnostics.

## Results

Data were analysed for 8042 HCWs that had logged door events and/or patient contacts in the UCLH Tower building between January 2020 and March 2021. This included both medical staff and non medical staff (cleaners, porters and admin). During the entire observation period 5,510,359 door events were recorded. In total, 21,801 patients were detected in the routinely collected data, of which 1707 (8%) were positive for COVID-19. Of the 6,931,878 patient contacts recorded, 1,643,113 (24%) were with COVID-19 patients. Table 2 provides a summary for the different stages of the pandemic.

In the following sections we describe the temporal and spatial patterns in the behaviour of HCWs, and how these changed throughout the pandemic. We also describe epidemiologically relevant changes in the patterns of spatial connectivity and indirect contacts between patients.

**Temporal dynamics.** During the baseline period, the daily number of door events showed clear temporal regularity, whereby the number of events was highest during weekdays (Fig. 1a). These peaks were in line with the daily staffing levels (Fig. 1b). The hourly number of door events were highest on weekdays between 7 am and 5 pm, but this peak was less prominent at weekends (Fig. 2a). The daily number of patient contacts did not exhibit an obvious weekly pattern, but the daily number of patients was highest during weekdays. Regardless of the day, the hourly number of patient contacts peaked once at 10 am and again at 6 pm (Fig. 2e). These temporal patterns demonstrate the utility of the routinely collected data in depicting staff and patient levels, in addition to the global activity of HCWs, which will underline the nature of staff working patterns within the hospital. Below we describe how staffing levels, patient numbers, HCW mobility (Fig. 2b–d) and patient contacts (Fig. 2f–h) changed during the pandemic compared to pre-pandemic levels.

**Table 2 Summary for metrics on healthcare worker behaviour derived from routinely collected data at the Tower building of University College London Hospital during the first year of the COVID-19 pandemic.**

|  | Baseline | First wave | Summer lull | Second wave |
|---|---|---|---|---|
|  | Jan–Feb 2020 | March–June 2020 | July–Aug 2020 | Nov 2020–March 2021 |
| Days | 61 | 123 | 91 | 147 |
| No. of staff \|$H$\| | 4801 | 5562 | 5158 | 6165 |
| No. of patients \|$P$\| | 6834 | 5853 | 5924 | 8093 |
| No. of door events \|$M$\| | 795,057 | 1,598,378 | 1,095,257 | 2,021,667 |
| No. of patient contacts \|$C$\| | 1,085,400 | 1,660,182 | 1,390,642 | 2,795,654 |
| No. of COVID-19$^+$ patients \|$P^{Positive}$\| (% of all patients) | 37 (1%) | 567 (10%) | 64 (1%) | 1095 (14%) |
| No. of contacts with COVID-19$^+$ \|$C^{Positive}$\| (% of all contacts) | 37,789 (4%) | 539,546 (34%) | 43,608 (3%) | 1,022,170 (37%) |

For each stage of the pandemic (pre-pandemic baseline, first wave, summer lull and second wave), the number of observation days is reported along with the total number of healthcare workers, patients, patients positive for COVID-19 (COVID-19$^+$), door events, patient contacts, and contacts with COVID-19$^+$ patients. For counts involving COVID-19$^+$ patients, the percentage of all patients/contacts are provided in brackets.

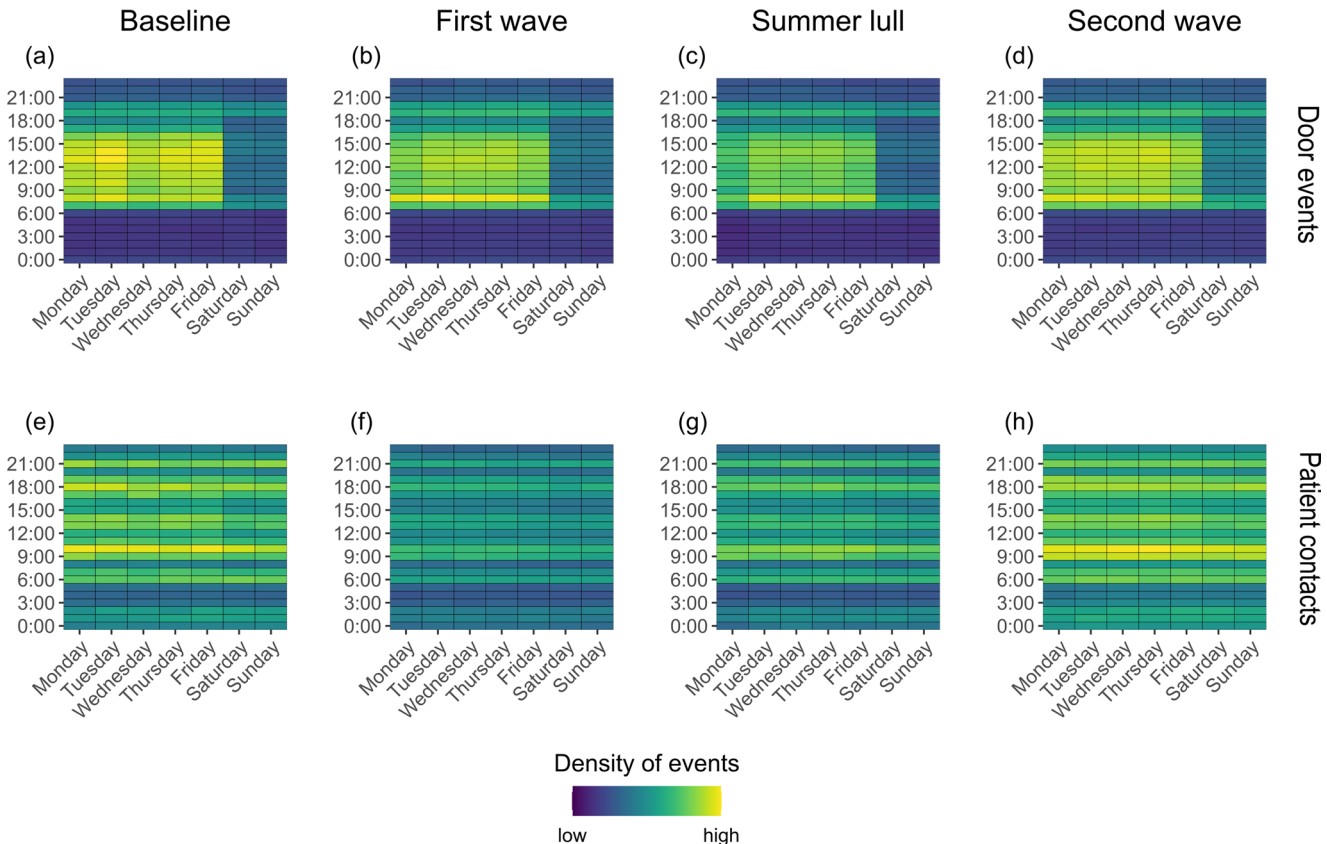

**Fig. 2 Heat maps for the hourly activity of healthcare workers in the Tower building of University College London Hospital during the first year of the COVID-19 pandemic.** The hourly density of events throughout the week are plotted for the number of door events (**a–d**) and number of patient contacts (**e–h**) during the baseline period (pre-pandemic), first wave, summer lull and second wave. Yellow cells represent a relatively high density of events and blue cells represent a relatively low density; to allow comparison between each stage of the pandemic, the colour gradient is relative to the maximum weight across all time periods for each metric respectively.

The first wave of COVID-19 patients was associated with a drop in weekday staffing levels (model estimate ($\hat{\beta}$) = −205; 95% confidence intervals (CI) = −273, −136; $t_{113}$ = −7.965; $p < 0.001$; $N_{Firstwave}^{H,Weekday}$ = −10%; Fig. 1b; Table 3), while no difference was observed during weekends ($\hat{\beta}$ = −9; CI = −55, 38; $t_{301}$ = −0.495; $p$ = 1.000). The daily number of door events was higher during weekends ($\hat{\beta}$ = 517; CI = 6, 1029; $t_{113}$ = 2.074; $p$ = 0.046; $N_{Firstwave}^{M,Weekend}$ = 6%), but remained consistent during weekdays ($\hat{\beta}$ = −354; CI = −903, 198; $t_{301}$ = −1.701; $p$ = 0.054) which is

surprising given the reduced staff levels. This is partly explained by an increase in the per HCW rate of daily door events during weekdays ($\hat{\beta}$ = 1.1; CI = 0.7, 1.4; $t_{301}$ = 8.072; $p < 0.001$; $N_{Firstwave}^{Mr,Weekday}$ = 11%) and weekends ($\hat{\beta}$ = 0.8; CI = 0.3, 1.4; $t_{113}$ = 3.936; $p$ = 0.001; $N_{Firstwave}^{Mr,Weekend}$ = 7%), suggesting that HCWs working during the first wave had higher levels of mobility than those pre-pandemic. The daily number of patients in the hospital reduced ($\hat{\beta}$ = −265; CI = −293, −236; $t_{418}$ = −25.849; $p < 0.001$; $N_{Firstwave}^{P}$ = −44%), and this coincided with a decrease in the daily

**Table 3 Average daily staffing levels, patient levels, rates of healthcare worker mobility and rates of patient contact in the Tower building at University College London Hospital during the first year of the COVID-19 pandemic.**

| | Baseline | First wave | Summer lull | Second wave |
|---|---|---|---|---|
| | Jan–Feb 2020 | March–June 2020 | July–Aug 2020 | Nov 2020–March 2021 |
| Weekday staffing levels | 1648 (1599–1698) | 1444*** (1411–1476) | 1414*** (1392–1437) | 1495*** (1468–1521) |
| Weekend staffing levels | 768 (758–779) | 760 (743–777) | 714* (704–724) | 856*** (828–884) |
| Patient levels | 598 (585–611) | 333*** (314–353) | 407*** (399–415) | 439*** (433–445) |
| No. of weekday door events | 14,890 (14,522–15,259) | 14,538 (14,319-14,756) | 13,419*** (13,101–13,737) | 15,163 (14,969–15,357) |
| No. of weekend door events | 8599 (8423–8775) | 9116* (8877–9355) | 8174 (7999–8350) | 9982*** (9720–10,243) |
| Rate of weekday door events | 9.1 (8.9–9.2) | 10.1*** (10.0–10.3) | 9.5* (9.3–9.6) | 10.2*** (10.1–10.4) |
| Rate of weekend door events | 11.2 (11–11.4) | 12.0*** (11.8–12.2) | 11.4 (11.3–11.6) | 11.7 (11.4–12.0) |
| No. of patient contacts | 17793 (17,540–18,047) | 13,497*** (13,142–13,853) | 15,282*** (15,023–15,541) | 19,018*** (18,719–19,317) |
| Rate of patient contact | 29.9 (29.4–30.3) | 42.6*** (41.2–44.0) | 37.7*** (37.2–38.1) | 43.6*** (42.7–44.4) |

The mean and 95% confidence intervals are reported for each metric during the pre-pandemic (baseline) period and each stage of the pandemic (first wave, summer lull and second wave). Linear models were used to determine if metrics for each stage of the pandemic were significantly different to baseline levels.
*$p < 0.05$, **$p < 0.01$, ***$p < 0.001$.

number of patient contacts logged by HCWs ($\hat{\beta} = -4296$; CI $= -4991$, $-3601$; $t_{418} = -16.387$; $p < 0.001$; $N^{C}_{Firstwave} = -24\%$), which was associated with a less prominent pattern in the hourly counts of contacts (Fig. 2f). However, the per patient rate of daily contact events was higher than at baseline ($\hat{\beta} = 12.8$; CI $= 10.5$, $15.0$; $t_{418} = 15.106$; $p < 0.001$; $N^{Cr}_{Firstwave} = 42\%$), suggesting HCWs had more contact events per patient than that logged pre-pandemic.

In the summer lull, when there were fewer COVID-19 patients in the hospital, daily patient numbers remained lower than baseline levels ($\hat{\beta} = -191$; CI $= -221$, $-161$; $t_{418} = -16.967$; $p < 0.001$; $N^{P}_{Summerlull} = -32\%$), as did staffing levels during weekdays ($\hat{\beta} = -234$; CI $= -306$, $-162$; $t_{301} = -8.671$; $p < 0.001$; $N^{H,Weekday}_{Summerlull} = -14\%$) and weekends ($\hat{\beta} = -54$; CI $= -104$, $-4$; $t_{113} = -2.916$; $p = 0.026$; $N^{H,Weekend}_{Summerlull} = -7\%$). While the daily and hourly pattern of patient contacts began to return towards that seen pre-pandemic, the daily count of events remained lower than during the baseline ($\hat{\beta} = -2512$; CI $= -3246$, $-1777$; $t_{418} = 9.067$; $p < 0.001$; $N^{C}_{Summerlull} = -14\%$), and the daily rate of contact was maintained above baseline levels ($\hat{\beta} = 7.8$; CI $= 5.4$, $10.2$; $t_{418} = 8.727$; $p < 0.001$; $N^{Cr}_{Summerlull} = 26\%$). During the weekdays, the daily number of door events were lower than baseline levels ($\hat{\beta} = -1471$; CI $= -2049$, $-893$; $t_{301} = -6.760$; $p < 0.001$; $N^{M,Weekday}_{Summerlull} = -10\%$) and the rate of mobility was higher on average ($\hat{\beta} = 0.4$; CI $= 0.1$, $0.8$; $t_{301} = 3.003$; $p = 0.017$; $N^{Mr,Weekday}_{Summerlull} = 4\%$), while no difference in either was observed during weekends (M: $\hat{\beta} = -424$; CI $= -974$, $125$; $t_{113} = -2.074$; $p = 0.242$; Mr: $\hat{\beta} = 0.3$; CI $= -0.3$, $0.9$; $t_{113} = 1.134$; $p = 1.000$).

During the second wave, the daily number of patients in the hospital remained lower than that at baseline ($\hat{\beta} = -159$; CI $= -186$, $-131$; $t_{418} = 8.671$; $p < 0.001$; $N^{P}_{Secondwave} = -27\%$), but an increase was observed in the daily number ($\hat{\beta} = 1225$; CI $= 549$, $1901$; $t_{418} = -4.803$; $p < 0.001$; $N^{C}_{Secondwave} = 7\%$) and rate of patient contacts ($\hat{\beta} = 13.7$; CI $= 11.5$, $15.9$; $t_{418} = 16.676$; $p < 0.001$; $N^{Cr}_{Secondwave} = 46\%$). Weekday staff numbers remained lower than baseline levels ($\hat{\beta} = -154$; CI $= -220$, $-87$; t$_{301} = -6.162$; $p < 0.001$; $N^{H,Weekday}_{Secondwave} = -9\%$), but had increased during weekends ($\hat{\beta} = 88$; CI $= 43$, $133$; $t_{113} = 5.195$; $p < 0.001$;

$N^{H,Weekend}_{Secondwave} = 11\%$). It is worth noting how daily staffing levels, after an initial drop during the Christmas break, followed the rise and fall of COVID-19 patients, emphasising a different strategy by the hospital to that in the first wave; where staff numbers were reduced. The daily number of door events only increased during weekends ($\hat{\beta} = 1383$; CI $= 883$, $1883$; $t_{113} = 7.425$; $p < 0.001$; $N^{M,Weekend}_{Secondwave} = 16\%$), while the rate of mobility increased only on weekdays ($\hat{\beta} = 1.1$; CI $= 0.8$, $1.5$; $t_{301} = 8.939$; $p < 0.001$; $N^{Mr,Weekday}_{Secondwave} = 12\%$).

**Spatial-temporal dynamics.** Overall, during the first and second waves, changes in the daily rate of HCW mobility ($N^{Mr}_{t}$) and patient contacts ($N^{Cr}_{t}$) were statistically higher on COVID-19 wards than non-COVID-19 wards, with no significant difference during the summer lull (Supplementary Table S1). However, considerable spatial variation was observed in the response from HCW mobility and patient contacts to the different stages of the pandemic. Here we focus on the behaviour of HCWs on the six floors that handled the majority ($> = 15\%$) of COVID-19 patients; AMU (floor 1), critical care (floor 3), HASU (floor 7), respiratory diseases (floor 8), general surgery (floor 9) and CoE (floor 10). We report the results for non-COVID-19 floors in Supplementary Table S2 and Table S3. It is worth noting that the emergency department (ground floor) experienced a large number of COVID-19 patients (Supplementary Fig. S1) but was not considered a COVID-19 floor; patients were triaged and then moved to a relevant ward.

With the exception of AMU, $N^{Mr}_{t}$ during the first wave was higher on all COVID-19 floors compared to baseline levels (Table 4; Fig. 3a–f). Despite this, on floors with HASU, general surgery and CoE, there was a negative relationship between $N^{Mr}_{t}$ and the number of COVID-19 patients on these floors (Table 5). In contrast, $N^{Mr}_{t}$ increased with the number of COVID-19 patients on AMU, critical care and the respiratory ward. During the summer lull, $N^{Mr}_{t}$ was no different to baseline levels on the floor with general surgery, but was lower on AMU and higher on all other COVID-19 floors; the most notable increase was on HASU ($N^{Mr}_{Floor7} = 101\%$). During the summer lull, a positive relationship was observed between $N^{Mr}_{t}$ and the number of COVID-19 patients on floors with HASU and CoE. In response to the second wave of COVID-19 patients, $N^{Mr}_{t}$ increased above

**Table 4 Changes in the rate of healthcare worker mobility and patient contacts on COVID-19 floors in the Tower building at University College London Hospital during the first year of the COVID-19 pandemic.**

| Floor | Mobility | | | | Patient contacts | | | |
|---|---|---|---|---|---|---|---|---|
| | Baseline | First wave | Summer lull | Second wave | Baseline | First wave | Summer lull | Second wave |
| AMU (Floor 1) | 5.8 (5.6, 6.0) | 7% (−4%, 18%) | −14%** (−26%, −3%) | −3% (−13%, 8%) | 19.6 (18.9, 20.3) | 45%*** (22%, 68%) | 35%*** (11%, 59%) | 58%*** (36%, 80%) |
| Critical Care (Floor 3) | 4.2 (4.0, 4.3) | 25%*** (14%, 36%) | 16%** (4%, 27%) | −3% (−14%, 7%) | 98.6 (95.5, 101.8) | 28%** (5%, 50%) | 16% (−8%, 40%) | 20% (−2%, 42%) |
| HASU (Floor 7) | 2.5 (2.4, 2.6) | 73%*** (62%, 84%) | 101%*** (90%, 112%) | 78%*** (68%, 88%) | 34.6 (33.8, 35.5) | 73%*** (50%, 96%) | −21% (−45%, 3%) | 155%*** (133%, 177%) |
| Respiratory/ Infection (Floor 8) | 7.1 (6.7, 7.4) | 14%** (3%, 25%) | 13%* (2%, 24%) | 66%*** (56%, 77%) | 23.5 (22.6, 24.5) | 14% (−9%, 36%) | 7% (−17%, 31%) | 46%*** (24%, 68%) |
| General Surgery (Floor 9) | 3.4 (3.3, 3.5) | 21%*** (10%, 32%) | −5% (−16%, 7%) | 50%*** (39%, 60%) | 24.3 (23.5, 25.2) | −7% (−30%, 16%) | 13% (−11%, 37%) | 26%** (4%, 48%) |
| CoE (Floor 10) | 3.1 (3.0, 3.2) | 60%*** (49%, 70%) | 44%*** (32%, 55%) | 66%*** (56%, 77%) | 24.8 (24.0, 25.5) | −2% (−24%, 21%) | 53%*** (29%, 77%) | 44%*** (22%, 66%) |

The average daily rates of healthcare worker mobility and patient contacts are reported for the pre-pandemic (baseline) period with 95% confidence intervals in brackets. Model estimates & 95% confidence intervals for the normalized difference to baseline are reported from linear models for daily rates of healthcare worker mobility ($N_t^{Mr}$) and patient contacts ($N_t^{Cr}$) during each stage of the pandemic (First wave, Summer lull and Second wave). COVD-19 floors are those that handled the majority of COVID-19 patients (>=15%) during the pandemic.
*p < 0.05, **p < 0.01, ***p < 0.001.

baseline levels on all COVID-19 floors, with the exception of AMU and critical care. $N_t^{Mr}$ had a positive association with the number of COVID-19 patients on all COVID-19 floors, with the exception of HASU where a negative relationship was observed.

During the first wave and compared to pre-pandemic levels, $N_t^{Cr}$ increased on AMU, critical care and HASU. There was a positive relationship between $N_t^{Cr}$ and the number of COVID-19 patients on floors with HASU, general surgery and CoE. During the summer lull, there was only an increase in $N_t^{Cr}$ on floors with AMU and CoE. $N_t^{Cr}$ was not correlated with the number of COVID-19 patients on any COVID-19 floor during the summer lull. With the exception of critical care, $N_t^{Cr}$ was higher on all COVID-19 floors during the second wave compared to pre-pandemic levels; with the most notable increase on HASU ($N_{Floor7}^{Cr} = 155\%$). A positive association was observed between $N_t^{Cr}$ and the number of COVID-19 patients on floors with AMU and HASU.

**Spatial connectivity.** The connectivity between floors (based on the number of HCWs that had activity on any two floors in the same day) revealed that some were more connected than others, and that the resulting clustering of floors varied throughout the pandemic (Fig. 4). Below we describe spatial clusters determined by the Louvain algorithm. Clusters identified using the leading eigenvector and random walk algorithms were similar and are reported in Supplementary Fig. S2.

During the baseline period, three clusters were identified; one large cluster (B1) containing Imaging through to Plant (floors −1 to 4), Short stay surgery (floor 6), HASU (floor 7) and General surgery (floor 9); a smaller cluster (B2) comprising the Basement (floor −2), Nuclear medicine (floor 5), Paediatrics (floor 11) and Adolescents (floor 12); and a third (B3) consisting of Respiratory disease (floor 8), CoE (floor 10) and Oncology through to Haematology (floors 13–16).

During the first wave, the connectivity between floors changed such that four clusters were identifiable, and floors adjacent to each other were generally in the same cluster. The basement through to Nuclear medicine (floors −2 to 5) formed one cluster (FW1), which included two COVID-19 floors (floors 1 & 3). Short stay surgery through to CoE (floors 6 to 10) formed a second cluster (FW2), all of which had COVID-19 patients, but only floor 6 was considered a non-COVID-19 floor. Paediatrics and adolescents (floors 11 and 12) made up a third cluster (FW3) and the fourth cluster (FW4) consisted of Oncology through to Haematology (floors 13 to 16); neither cluster contained floors that handled the majority of COVID-19 patients.

During the summer lull only three clusters were identified. Imaging through to CoE (floors −1 to 10) formed the largest cluster (SL1), and this included all floors identified as COVID-19 floors. The basement, Paediatrics and Adolescents (floors −2, 11 and 12) were in a second cluster (SL2), and the third cluster (SL3) consisted of Oncology through to Haematology (floors 13 to 16). During the second wave the connectivity of floors and the clusters they formed were unchanged from that in the summer lull, suggesting that the spatial activity of HCWs had stabilised.

**Indirect contacts between patients.** The average daily connectivity between COVID-19 negative patients (due to shared contacts with HCWs on the same day; $S_t^{Negative}$; Fig. 5) remained stable, on average, throughout the pandemic ($\bar{S}_{Firstwave}^{Negative} = 5\%$; $\bar{S}_{Summerlull}^{Negative} = 5\%$; $\bar{S}_{Secondwave}^{Negative} = 5\%$). However, $S_t^{Negative}$ increased by 0.30% (CI = 0.09%, 0.51%; $t_{355} = 2.786$; $p = 0.006$) for every two fold increase in the number of COVID-19 patients in hospital during the first wave. No significant effect was found during the summer lull ($\hat{\beta} = -0.21\%$; CI = −0.56%, 0.15%; $t_{355} = -1.154$;

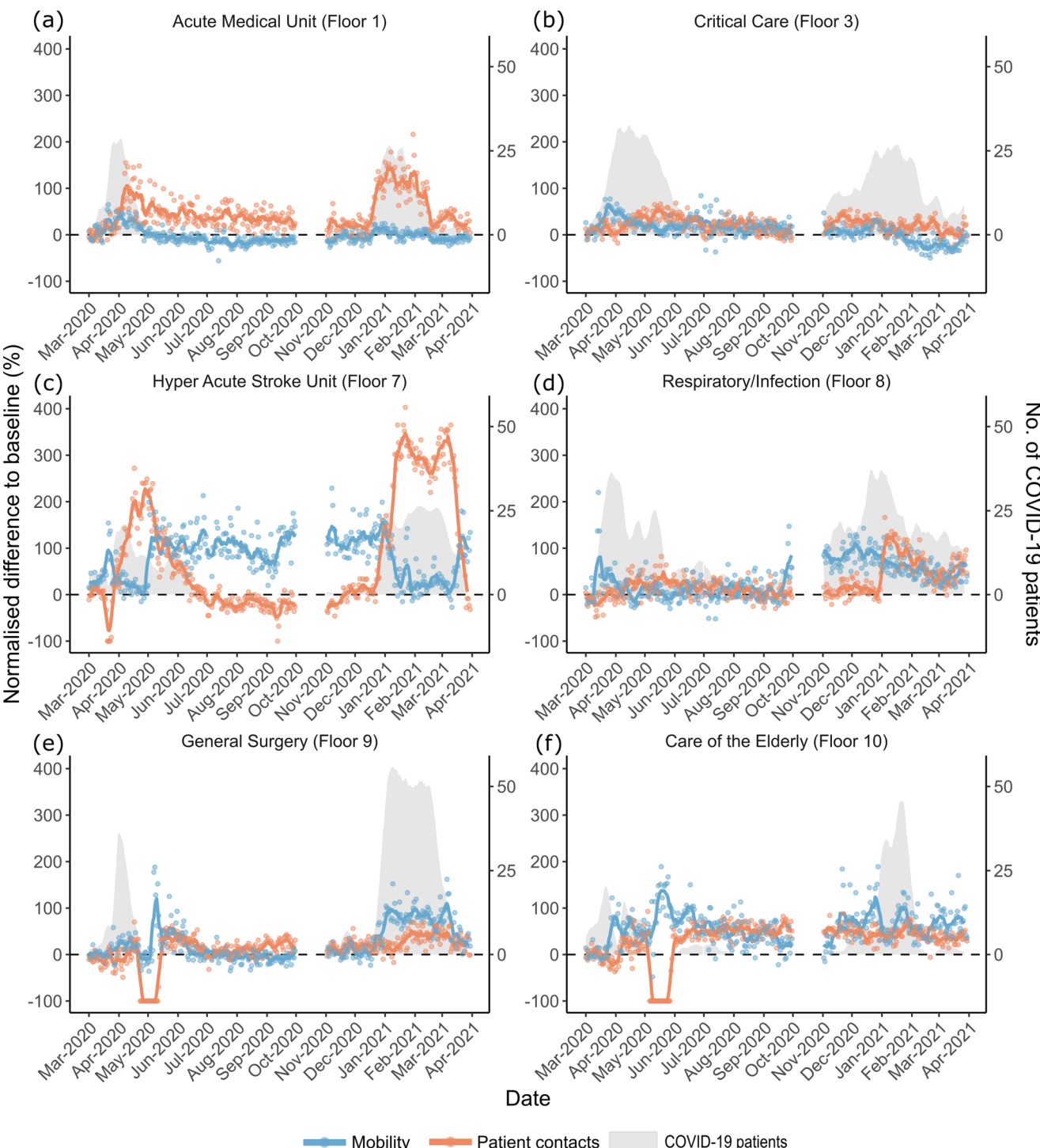

**Fig. 3 Changes in the daily rate of healthcare worker mobility and patient contacts on COVID-19 floors in the Tower building at University College London Hospital during the first year of the COVID-19 pandemic.** The normalized difference to baseline for daily rates of healthcare worker mobility ($N_t^{Mr}$; blue points) and patient contacts ($N_t^{Cr}$; red points) are plotted for COVID-19 floors: **a** Acute medical unit. **b** Critical care. **c** Hyper acute stroke unit. **d** Respiratory/infection ward. **e** General surgery. **f** Care of the elderly. The solid blue and red lines represent the seven day rolling averages. The horizontal black dotted dashed line represents 0% change compared to the average in the 2 months pre-pandemic. COVID-19 wards were identified as those that handled the majority (>= 15%) of all COVID-19 patients in the hospital during the observation period. The grey polygons indicate the number of COVID-19 patients on the floor. Data for October 2020 was not available.

$p = 0.249$) or during the second wave ($\hat{\beta} = 0.20\%$; CI = 0.00%, 0.41%; $t_{355} = 1.922$; p = 0.055).

In contrast, the hospital reduced the daily connectivity between COVID-19 negative and positive patients ($S_t^{Positive}$) during the first and second waves to a low of < 1% ($\bar{S}_{Firstwave}^{Positive} = 2\%$; $\bar{S}_{Secondwave}^{Positive} = 2\%$). However, this was not maintained during the summer lull ($\bar{S}_{Summerlull}^{Positive} = 4\%$) suggesting a relaxation in staff cohorting procedures. These patterns highlight a reactive

**Table 5 The relationship between the number of COVID-19 patients in the hospital and HCW mobility and patient contacts on COVID-19 floors in the Tower building at University College London Hospital during the first year of the COVID-19 pandemic.**

| Floor | Mobility | | | Patient contacts | | |
|---|---|---|---|---|---|---|
| | First wave | Summer lull | Second wave | First wave | Summer lull | Second wave |
| AMU (Floor 1) | 9%*** (6%, 12%) | 0% (−13%, 13%) | 3%** (1%, 6%) | 2% (−4%, 7%) | 0% (−27%, 27%) | 25%*** (20%, 30%) |
| Critical Care (Floor 3) | 4%* (0%, 7%) | −1% (−10%, 9%) | 13%*** (7%, 18%) | 3% (−5%, 10%) | −3% (−22%, 17%) | 4% (−7%, 16%) |
| HASU (Floor 7) | −22%*** (−26%, −19%) | 26%*** (13%, 38%) | −18%*** (−20%, −16%) | 37%*** (30%, 44%) | 4% (−23%, 30%) | 69%*** (64%, 73%) |
| Respiratory/Infection (Floor 8) | 8%*** (4%, 12%) | 4% (−3%, 12%) | 17%*** (8%, 25%) | 0% (−8%, 8%) | 3% (−12%, 18%) | 6% (−12%, 25%) |
| General Surgery (Floor 9) | −4%* (−6%, −1%) | 7% (−6%, 21%) | 19%*** (16%, 21%) | 8%** (2%, 13%) | −13% (−41%, 15%) | 4% (−1%, 9%) |
| CoE (Floor 10) | −18%*** (−21%, −15%) | 10%** (3%, 17%) | 3%** (1%, 5%) | 17%*** (11%, 23%) | −4% (−18%, 11%) | 1% (−4%, 5%) |

Model estimates & 95% confidence intervals for the change in the normalized difference to baseline are reported from linear models for daily rates of healthcare worker mobility ($N_t^{Mr}$) and patient contacts ($N_t^{Pc}$). The number of COVID-19 patients in hospital was included as a logged (base 2) term in models, and estimates should therefore be interpreted as the change in $N_t^{Mr}$ or $N_t^{Pc}$ with every doubling in the number of COVID-19 patients in the hospital during each stage of the pandemic (First wave, Summer lull and Second wave). COVID-19 floors are those that handled the majority of COVID-19 patients (>= 15%) during the pandemic.
*p < 0.05, **p < 0.01, ***p < 0.001.

response to the rise of COVID-19 patients and is further supported whereby, for every doubling in the number of COVID-19 patients in the hospital, $S_t^{Positive}$ decreased by 0.81% (CI = 0.61%, 1.00%; $t_{355} = -8.188$; $p < 0.001$) during the first wave and by 0.56% (CI = 0.37%, 0.75%; $t_{355} = -5.767$; $p < 0.001$) during the second wave; no significant relationship was found during the summer lull ($\hat{\beta} = 0.25\%$; CI = −0.07%, 0.58%; $t_{355} = 1.529$; $p = 0.127$).

During the first wave a noteworthy spike occurred in $S_t^{Positive}$ to 16% on the 5th May 2020. This was due to one HCW who had contact with 87 patients, 32 of which were positive for COVID-19. Applied in real time, such insights could help practitioners quickly identify and address weaknesses in IPC activities that could compromise patient and staff safety.

## Discussion

Mobility and contact rates are fundamental to the transmission of communicable diseases, and data on these behaviours are extremely valuable for epidemiological investigations. It has long been established that HCWs can be part of transmission clusters within healthcare settings[13,16] however, data on the behaviour of HCWs are often scarce. In this paper, we demonstrate how behavioural markers for HCW mobility and patient contacts within the hospital, can be derived from EMRs and door access logs at an aggregate level. Using data from a London teaching hospital and during the COVID-19 pandemic, we provide a framework to further support IPC practitioners in assessing patterns of staff behaviour, identifying behavioural change and in conducting evidence-based infection control.

The temporal trends in workforce and HCW behaviour are in line with those reported in other studies[24,30,36–38]. Staff and patient levels determined daily patterns in the aggregate measures of HCW behaviour, and this was evident when the hospital reduced staff and patient numbers during the first wave of COVID-19 patients, which resulted in a notable drop in logged patient contacts. However, the rate of patient contact (number of contact events per patient) was maintained above baseline levels throughout the pandemic, as was the rate of HCW mobility (number of door events per HCW) during weekdays. Our framework illustrates the utility of the featured data sources in representing the working practices of HCWs, and their potential to passively monitor behaviour change and activity patterns of the HCW population. From an operational point of view this framework provides a means to quickly generate evidence of changing working practices and identify undesirable work pressures, and risk of workforce fatigue, and resulting illness and staff shortages.

Patterns of HCW behaviour showed considerable spatial-temporal variation in response to the pandemic. Increases in the rate of mobility and rate of patient contact were most notable on floors handling the majority of COVID-19 patients during the first and second waves, and we find evidence to suggest that on some floors the observed changes in behaviour were associated with shifts in the COVID-19 patient population. Exact causes for the observed changes in HCW behaviour are hard to ascertain, and are likely products of a combination of factors from shifting working practices (e.g. through IPC activities), perceptions of risk (e.g. before/after vaccination and changes in the availability of PPE) and hospital pressures (e.g. needs of the patient population). Furthermore, the degree of change in these behavioural markers was not equal across floors and, despite few (or no) COVID-19 patients, non-COVID-19 floors also experienced changes in staff behaviour. Differences in the trends of HCW behaviour on different floors will depend on the functions of the wards occupying them, how these functions evolved during the course of the

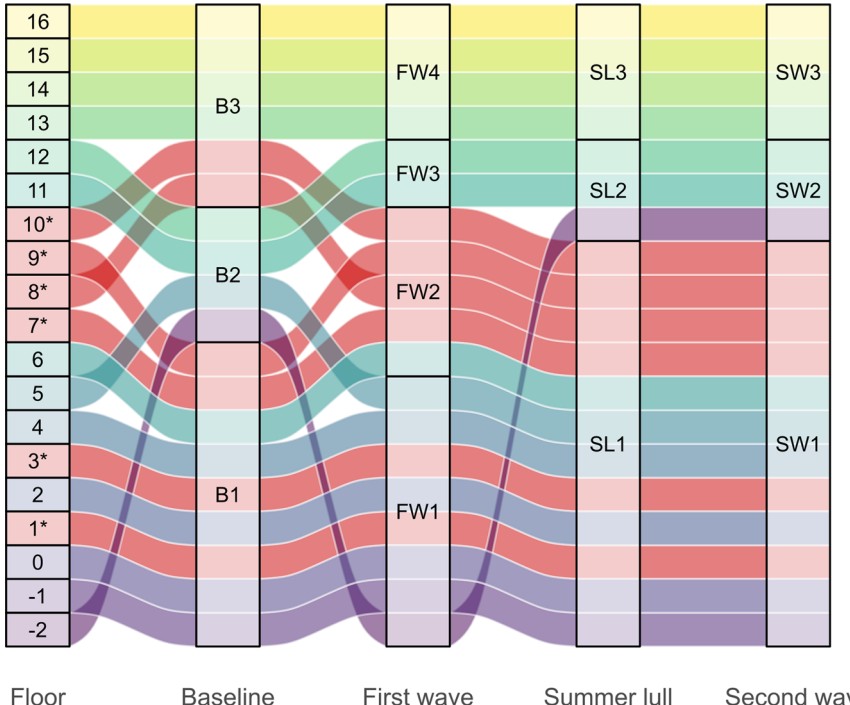

**Fig. 4 The spatial connectivity of floors in the Tower building at University College London Hospital during the first year of the COVID-19 pandemic.**
The alluvial diagram depicts the connectivity of floors, as determined from the Louvain clustering algorithm and using dyadic weights derived from the number of healthcare workers with door events and/or patient contacts on the focal floors during the same day. The numbering in the left most column identifies the floors, and the codes in the remaining four columns represent the cluster group the floors belong to in each stage of the pandemic; pre-pandemic (baseline; B), first wave (FW), summer lull (SL) and second wave (SW). Floors that handled the majority (> = 15%) of COVID-19 patients during the observation period are identified by an asterisk and their flows between clusters are in red. The flows for non COVID-19 floors are differentiated by colours ranging from yellow to blue.

pandemic, and on IPC interventions. Even where data on causative factors for observed behaviour change are not available, the framework provides insights to generate hypotheses and a means for further investigation.

One strategy to prevent nosocomial transmission is to cohort patients and staff, whereby patients positive for the disease of concern and/or the staff responsible for their care, are kept separate to the rest of the patient population[29]. At UCLH this was achieved by establishing COVID-19 wards that handled the majority of COVID-19 patients. Using the routinely-collected data we were able to identify the main COVID-19 wards and monitor the daily indirect contacts between patients (as determined through shared contacts with HCWs on the same day). Successful staff cohorting would have resulted in no indirect contact between COVID-19 negative and positive patients. However, this was not consistently achieved and, while the indirect contacts between these groups of patients were substantially reduced during the first and second waves, the response was not maintained during the summer lull, and appears reactive to increases in the number of COVID-19 patients. Staff cohorting can be prevented by numerous practical limitations, and the pandemic presented many challenges including staff shortages. Using EMRs to investigate indirect contacts between patients has been explored before[26], but this is not necessarily routine practice, and we provide such an analysis here to illustrate the diversity in applications for data on staff behaviour. It is evident that the routinely collected data provides a tool for IPC practitioners to monitor (in near to real time) the success of interventions such as cohorting, and offers a means to quickly identify, investigate and react to undesirable spikes in indirect patient contacts facilitated by HCWs that could compromise patient and staff safety.

Another strategy available to IPC practitioners is to limit the traffic within the hospital[29]. At UCLH a number of interventions were adopted to accomplish this, including reduced patient and staff numbers, and through the installation of COVID doors that created barriers to disrupt the flow of people between spaces. Our analyses identified the reduced staff and patient numbers, along with substantial changes in the connectivity of floors during the first wave, which then stabilised during the summer lull and second wave; indicating a new normal to working patterns. We were also particularly interested in the spatial connectivity of floors that handled the majority of COVID-19 patients with non-COVID-19 floors, as this could present opportunities for outbreaks. The flow of staff between floors throughout the pandemic was such that COVID-19 floors were closely associated with non-COVID-19 floors, therefore the flow of HCWs between areas with low and high burdens of the disease may have presented a risk of outbreaks if sufficient IPC measures were not in place. Had a framework like the one presented here been operationalised, decision-makers would have had a means to rapidly assess the spatial connectivity of spaces and use these data to justify further interventions/investigations to mitigate the associated risks. While our analysis on spatial connectivity provides an example of how data on HCW movements can support IPC, caution should be taken in the interpretation of results, as we were unable to assess the effect of COVID doors on the mobility of staff; due to missing information on the dates of their installation and use. A higher resolution analysis that takes into account the partitions within floors may reveal the true flow of staff between COVID-19 and non-COVID-19 areas.

In this investigation we used a minimal number of data fields and metrics aggregated at the level of the HCW population. However, further insights into the variations of HCW behaviour

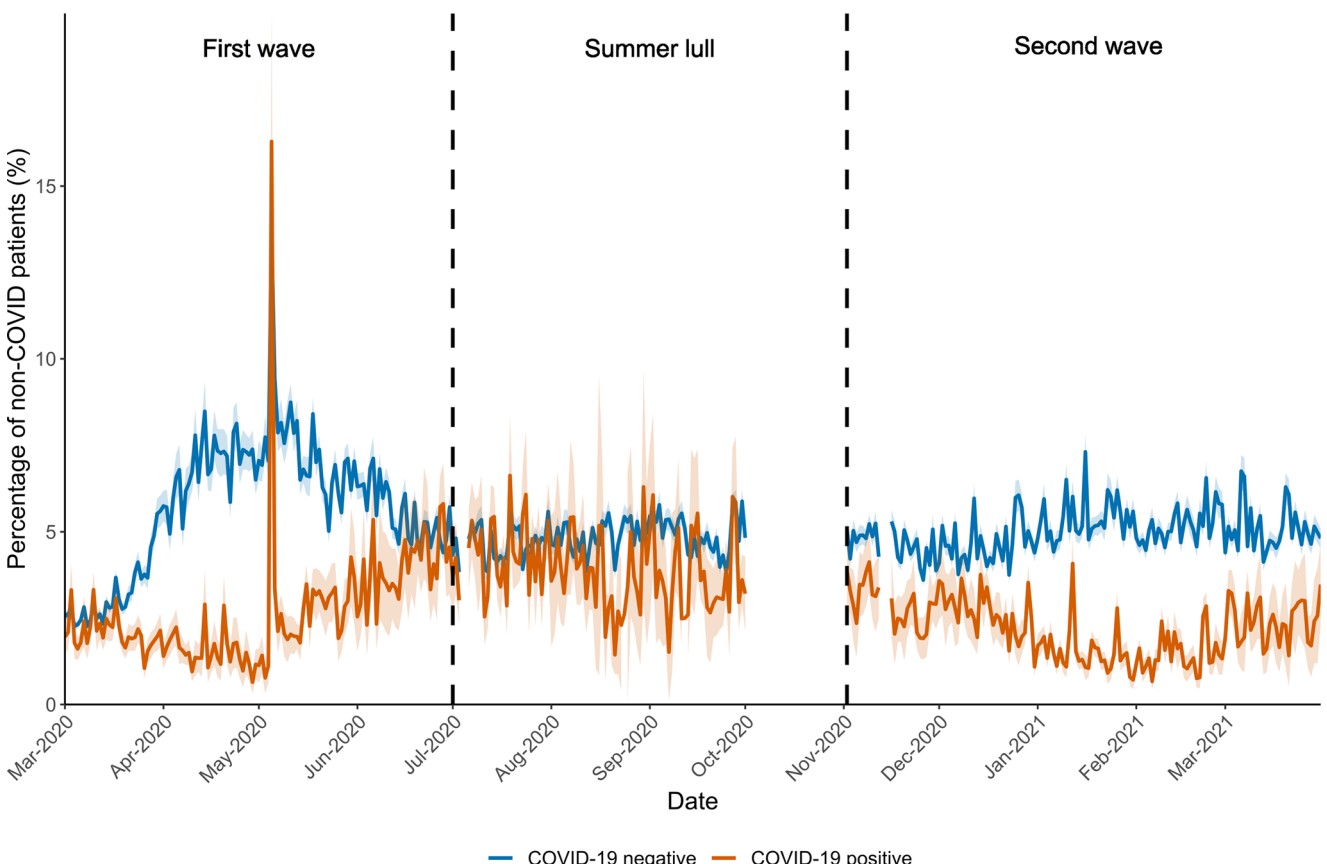

**Fig. 5 The average daily percentage of COVID-19 negative patients that had indirect contacts with other patients in the Tower building at University College London Hospital during the first year of the COVID-19 pandemic.** The daily averages and 95% confidence intervals are plotted separately for the percentage of COVID-19 negative patients to have indirect contacts with COVID-19 positive (orange line/shading) and other COVID-19 negative (blue line/shading) patients. An indirect contact was determined through evidence of a shared contact with healthcare workers on the same day. The vertical dashed black lines indicate the start/end points for each stage of the pandemic; first wave, summer lull and second wave. Data for October 2020 was not available.

could be uncovered if the data were paired with other data fields and aggregated by individual or HCW group. For example, combining this data with specific details on IPC activity, would allow investigations into the pre and post effects of interventions on HCW mobility and patient contacts. Combining these data sources with data from staff screening programmes could help identify HCW groups and individuals more at risk of acquiring HAIs, along with the behaviours or working conditions associated with higher risk. If shown to be epidemiologically relevant, the markers of HCW behaviour may also provide a dynamic tool to identify staff more at risk of involvement in chains of transmission, which could inform the targeted screening of staff. The data analysed here were anonymized and aggregated metrics used such that an individuals privacy was protected. That said, studies on HCW perceptions and acceptance on the use of passively collected data for routine surveillance are required to help address ethical concerns, and data should only be used for means of better protecting both staff and patients e.g. for informing IPC.

The framework outlined here provides a system wide perspective on staff behaviour that can enable exploration of specific and spatially discrete contexts. The measures facilitate comparisons between different occupational contexts, to generate and test hypotheses on behaviour change, and contribute to a better understanding of the spatial and temporal heterogeneity in the infection risk for HCWs; as was seen during the pandemic[15]. For this to be realised in routine practice, platforms such as data dashboards are required to enable exploitation of measures. We envision the resulting tools would be of use to practitioners to

facilitate rapid investigations, provide early warning systems and support decisions on policies and interventions, in addition to monitoring the effectiveness of such actions. Providing the digital infrastructure is in place, the framework can be adapted for use outside of the healthcare environment, such as contexts involving contact with wildlife or livestock where there is a risk of disease emergence. The development of digital systems for real-time behavioural monitoring related to disease transmission will contribute towards improved pandemic preparedness.

While the data sources featured here have potential to be used operationally by IPC practitioners in real time, there are several challenges that hospitals may have to overcome for this to be realised. Firstly, it is worth noting that the framework presented here relies on electronic records and, while UCLH is a digital hospital, many healthcare facilities in the UK and across the world are not. Hospitals, particularly those within the NHS, often outsource services such as systems for security door logs and EMRs, and in this study the various datasets from outsourced companies had to be consolidated, which required the creation of a master staff index to establish links between the databases. Mapping the data flows and creating a user friendly platform (such as a data dashboard) will be challenging, requiring the collaboration of researchers, IT professionals and IPC staff. There are also challenges in relation to the validation of these data, as we lack assurances on the exact nature of the processes underlying their generation. For instance, the use of staff cards to open security doors may be biased in time and space by HCWs following each other through doors (e.g. during ward rounds), or by doors being

left open. Likewise, there has been little systematic analysis to date in relation to the accuracy of the spatial or temporal markers from EMRs, or the HCWs involved in events. While these remain important validation challenges, the principles underlying aggregate patterns produced using these data appear sound.

In conclusion, this paper has described a framework to produce simple markers for the behaviour of staff in the healthcare environment from routinely collected data. Data on HCW behaviours are often scarce but, as hospitals embrace the digital age, data is becoming more readily available. Our framework provides a means to rapidly assess working patterns, investigate behaviour change and support evidence-based IPC activities in near to real-time. The integration of such frameworks into routine practice will be pivotal in building more resilient healthcare systems to better protect HCWs and patients, and to improve pandemic preparedness.

**Reporting summary**. Further information on research design is available in the Nature Portfolio Reporting Summary linked to this article.

## Data availability

The data are available from UCLH but restrictions apply to protect the privacy of individuals, so the data are not publicly available. Data are however available from the authors upon reasonable request and with permission from UCLH. The aggregated for data used to create Figs. 1, 3, and 5 are provided in Supplementary data 1.

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

## Acknowledgements

This study was supported by the UCLH/UCL NIHR Biomedical Research Centre and funding from the UKRI MRC (grant ref: MC_PC_19082), and UCLH Charity. For their support we thank the UCLH medical directors Charles House and Gill Gaskin, Pushpsen Joshi at the Joint Research Office, Nathan Lea from the UCLH information governance department, Leila Hail from the UCLH infection control department, Richard Clarke, David Ramlakhan, David Thompson and Gareth Adams at the UCLH digital services department, Keiran Suchak for methodological discussions, and all involved with the SAFER research programme.

## Author contributions

E.N., M.S., C.F.H., and E.M. supervised the project. E.N. and E.M. designed the research. W.W., C.L., and S.K. extracted the data. J.W.A., E.M., and N.G. analysed the data. J.W.A. and E.M. wrote the manuscript. All authors approved the final version.

## Competing interests

The authors declare no competing interests.
