## [Peer Review File · Communications Medicine]

This manuscript has been previously reviewed at another Nature Portfolio journal. This document only contains reviewer comments and rebuttal letters for versions considered at Communications Medicine

REVIEWERS' COMMENTS:

Reviewer #1 (Remarks to the Author):

The authors have addressed all my comments in a satisfactory way. The manuscript has undergone extensive changes that substantially strengthened the manuscript.

Reviewer #2 (Remarks to the Author):

Brief summary of the manuscript

This paper aims to mobilize passive data for use in hospital-based infection prevention and control efforts. It develops measures of mobility, patient contact, and social and spatial connectivity based on electronic medical records and security door access logs. It applies these measures to data from an academic medical facility during a pre-COVID baseline period and two waves of COVID infection and an intervening lull period. The resulting data evidence little by way of overarching trends though they contain some indications of reactive staff cohorting. The conclusions focus on potential for practical application.

Overall impression of the work

My overall impression of the work is of a paper that is highly lucid in the development and explication of indicators of HCW mobility and patient contacts that can be drawn from electronic health records and door access data. The collection of these measures is presented as a framework. Second, the paper applies these indicators to reveal changes in HCW mobility and indirect patient contacts over time periods defined with respect to the COVID-19 census in a single hospital. An additional aspect of the paper that makes a strong impression is that the focus on HCWs, which is novel and worthwhile.

Specific comments, with recommendations for addressing each comment

First, I thank the authors for their precise and clear development of useful measures. From my perspective, this is the great strength of this paper, and I hope to see the paper frequently cited and its measures enter into wider use.

The paper has been revised and improved substantially, and the prior reviewers appear to have made many helpful comments. This leaves to me one primary area.

The primary contribution of this paper is methodological development. However, when the measures are applied to the hospital in question, no theories or hypothesis on the expected relation of the measures to the hospital data are articulated in the methods (i.e., in advance of the analysis and results). Therefore, when the paper uses the measures developed, a comparatively universal approach is taken, and it is not surprising that several contra-intuitive or contradictory results thus emerge.

The paper addresses the presence of such results in a general way in the conclusion when stating (508-517): "However, exact causes for the observed changes in HCW behaviour are hard to ascertain, and are likely products of a combination of factors from shifting working practices (e.g. through IPC activities), perceptions of risk (e.g. before/after vaccination and changes in the availability of PPE) and hospital pressures (e.g. needs of the patient population). What's more, the degree of change in these behavioural markers was not equal across floors and, despite few (or no) COVID-19 patients, non COVID-19 floors also

experienced changes in staff behaviour. Differences in the trends of HCW behaviour on different floors will depend on the functions of the wards occupying them, how these functions evolved during the course of the pandemic and on IPC interventions.”

While I find the end-loading of explanation a little bit unsatisfying, in reading through the paper, the authors have used very precise wording that does not over-promise with respect to explanation. They authors lay greater emphasis on potential managerial application than on the development of middle-level theory in this paper. The large amount of results on the COVID phases and HCW movement and contacts notwithstanding, the paper delivers on its stated intents.

I have no specific comments with associated recommendations.

Dr. Julie Zook

Reviewer #3 (Remarks to the Author):

The first point I'd like to make is I like this paper, and I like what is being attempted here. I was not one of the original reviewers, so this is my impression of reading the revised manuscript as new. I do think it lacks a little in its structure and potential importance beyond a hospital data analytics audience. Continually saying how this would help epidemiologists without too much additional detail doesn't help. This could be improved upon. However, I also understand if other previous and current reviewers are happy, and believe this makes a contribution, then I'm happy to follow their lead.

For my perspective.

This is an important topic, to be able to (in near real time) understand infection / harm etc can certainly improve patient care.

Overall, I was impressed by the methods employed by the authors to make some sense of these data. Where I felt the paper was not as strong as I hoped was in conveying why this is important. I understand the importance and potential of different digital inputs, including EMR. However, while the authors spent a lot of time explaining the methodological steps, and then presenting results, and arguably this is needed. The “so what” factor was never removed. A few examples where a little more clarity would have helped was in the figures:

Figure 1 – figure would be better if you removed the daily low.

Figure 2 – I understand heat maps, but maybe a for example, and then select a cell and explain it.

Figure 3 – Really hard to understand what is happening

While a limited data analytics audience might not have as much problem with these results, I wonder if shifting some of the method details, and /or results to an addendum would help? There were some interpretations within the presentation of results, but I felt they were lost in the overwhelming descriptions. Maybe a little more attention in the discussion stage explaining the key findings that go beyond what would expect from the situation. Why is all

this work illuminating. Especially as the authors describe some of the data and technical challenges involved. What is the real benefit of taking all this on?

I agree about how this could be useful in some form of real time information system – but no real detail is presented as to how that could be achieved – even conceptually.

So I'm left a little flat – I admire the topic, I do believe in its overall value, and I am impressed with the methods and application of the authors. But I'm not sure anyone reading the manuscript currently would be inspired to change operations.

Response to reviewers comments

Manuscript ID: COMMSMED-22-0340-T

Title: Investigating healthcare worker mobility and patient contacts within a UK hospital during the COVID-19 pandemic

Authors: Jared K Wilson-Aggarwal, Nick Gotts, Wai Keong Wong, Chris Liddington, Simon Knight, Moira J Spyer, Catherine Houlihan, Eleni Nastouli, Ed Manley

We would like to thank the reviewers for taking the time to read and comment on our manuscript. We have revised the manuscript to address any reviewer concerns/suggestions and to comply with editorial requests. We have also re-run all counts and statistics, carefully checking rounding and final numbers.

Below we have provided a response to each reviewer comment and refer to lines in the revised manuscript (in simple markup where tracked changes are present). Reviewers' comments are in *italics* and our response is in **bold**.

Reviewer #1 (Remarks to the Author):

The authors have addressed all my comments in a satisfactory way. The manuscript has undergone extensive changes that substantially strengthened the manuscript.

We'd like to thank the reviewer for their comments which helped improve the previous version of the manuscript.

Reviewer #2 (Remarks to the Author):

Brief summary of the manuscript

This paper aims to mobilize passive data for use in hospital-based infection prevention and control efforts. It develops measures of mobility, patient contact, and social and spatial connectivity based on electronic medical records and security door access logs. It applies these measures to data from an academic medical facility during a pre-COVID baseline period and two waves of COVID infection and an intervening lull period. The resulting data evidence little by way of overarching trends though they contain some indications of reactive staff cohorting. The conclusions focus on potential for practical application.

Overall impression of the work

My overall impression of the work is of a paper that is highly lucid in the development and explication of indicators of HCW mobility and patient contacts that can be drawn from electronic health records and door access data. The collection of these measures is presented as a framework. Second, the paper applies these indicators to reveal changes in HCW mobility and indirect patient contacts over time periods defined with respect to the COVID-19 census in a single hospital. An additional aspect of the paper that makes a strong impression is that the focus on HCWs, which is novel and worthwhile.

Specific comments, with recommendations for addressing each comment

First, I thank the authors for their precise and clear development of useful measures. From my perspective, this is the great strength of this paper, and I hope to see the paper frequently cited and its measures enter into wider use.

Thank you, we are happy to hear that the reviewer finds our work clear and comprehensible, and that they agree the focus on HCWs is worthwhile and novel.

The paper has been revised and improved substantially, and the prior reviewers appear to have made many helpful comments. This leaves to me one primary area.

The primary contribution of this paper is methodological development. However, when the measures are applied to the hospital in question, no theories or hypothesis on the expected relation of the measures to the hospital data are articulated in the methods (i.e., in advance of the analysis and results). Therefore, when the paper uses the measures developed, a comparatively universal approach is taken, and it is not surprising that several contra-intuitive or contradictory results thus emerge.

We agree that hypotheses were lacking, and we have now addressed this in the final paragraph of the introduction (L115-123). In addition to the two primary aims of the paper, we now state the hypotheses that motivated this manuscript, with the expectation that the number of COVID-19 patients in the hospital will be associated with any changes in the behavioural metrics.

The paper addresses the presence of such results in a general way in the conclusion when stating (508-517): “However, exact causes for the observed changes in HCW behaviour are hard to ascertain, and are likely products of a combination of factors from shifting working practices (e.g. through IPC activities), perceptions of risk (e.g. before/after vaccination and changes in the availability of PPE) and hospital pressures (e.g. needs of the patient population). What’s more, the degree of change in these behavioural markers was not equal across floors and, despite few (or no) COVID-19 patients, non COVID-19 floors also experienced changes in staff behaviour. Differences in the trends of HCW behaviour on different floors will depend on the functions of the wards occupying them, how these functions evolved during the course of the pandemic and on IPC interventions.”

While I find the end-loading of explanation a little bit unsatisfying, in reading through the paper, the authors have used very precise wording that does not over-promise with respect to explanation. They authors lay greater emphasis on potential managerial application than on the development of middle-level theory in this paper. The large amount of results on the COVID phases and HCW movement and contacts notwithstanding, the paper delivers on its stated intents.

I have no specific comments with associated recommendations.

Dr. Julie Zook

We thank the reviewer for their comments and for recognizing the care taken not to over embellish the results. We believe there is a great potential for these data sources to support investigations into mid-level theory, especially those relating to IPC, and hope our framework will provide a foundation for further investigations.

We accept the critique of the ‘end-loaded’ explanation, and ideally we would have liked access to additional data sources to carry the analysis beyond indicative findings. However, as the reviewer points out, the manuscript delivers in its main aim of outlining a framework to measure HCW behaviour in the healthcare environment. Furthermore, while we are unable to determine causative factors for behaviour change, the framework provides a means to observe data,

generate hypotheses and then conduct investigations, which we briefly discuss in the discussion (L470-471).

Reviewer #3 (Remarks to the Author):

The first point I'd like to make is I like this paper, and I like what is being attempted here. I was not one of the original reviewers, so this is my impression of reading the revised manuscript as new. I do think it lacks a little in its structure and potential importance beyond a hospital data analytics audience. Continually saying how this would help epidemiologists without too much additional detail doesn't help. This could be improved upon. However, I also understand if other previous and current reviewers are happy, and believe this makes a contribution, then I'm happy to follow their lead.

Thank you, we are glad the reviewer enjoyed the manuscript and appreciates the aim of this work. We have addressed the concerns raised and detail these in our replies to the comments below.

For my perspective.

This is an important topic, to be able to (in near real time) understand infection / harm etc can certainly improve patient care.

Overall, I was impressed by the methods employed by the authors to make some sense of these data. Where I felt the paper was not as strong as I hoped was in conveying why this is important. I understand the importance and potential of different digital inputs, including EMR. However, while the authors spent a lot of time explaining the methodological steps, and then presenting results, and arguably this is needed. The "so what" factor was never removed. A few examples where a little more clarity would have helped was in the figures:

Figure 1 – figure would be better if you removed the daily low.

Thank you for this suggestion. We have now edited this figure considerably to make it more attractive and to meet editorial requests. We have kept the daily lows as these depict the reduced activity at weekends which are an important temporal feature of the data and help to highlight the utility of these data sources in representing HCW working patterns.

Figure 2 – I understand heat maps, but maybe a for example, and then select a cell and explain it.

For clarity we have edited this figure and, rather than providing an example, we now include a legend to allow the reader to quickly interpret the colours, and have added borders around the hourly cells to help better differentiate each.

Figure 3 – Really hard to understand what is happening

This figure has been edited to improve the clarity of the visualisation.

While a limited data analytics audience might not have as much problem with these results, I wonder if shifting some of the method details, and /or results to an addendum would help? There were some interpretations within the presentation of results, but I felt they were lost in the overwhelming descriptions. Maybe a little more attention in the discussion stage explaining the key findings that go beyond what would expect from the situation. Why is all this work illuminating. Especially as the authors describe some of the data and technical challenges involved. What is the real benefit of taking all this on?

Thank you for these suggestions and for raising the concern relating to the need to address the ‘so what’ and ensure the value of the paper goes beyond the data analytics audience. While those interested in data analytics are a key audience, we also hope our work sparks the interest of IPC practitioners and others with an interest in safeguarding staff and patients.

We have now edited the discussion to help engage a wider audience and emphasis the benefit of adopting such a framework. Specifically, we provide an additional example of how this data might be used (to evidence undesirable working pressures of HCWs; L452-457), explain how we envision this framework might be used (L530-532) and mention how this framework might be applied outside of the healthcare environment (L533-538). Finally, we summarise the benefit of adopting such a framework in the concluding paragraph of the discussion ‘...Data on HCW behaviours are often scarce but... Our framework provides a means to rapidly assess working patterns, investigate behaviour change and support evidence-based IPC activities in near to real time. The integration of such frameworks into routine practice will be pivotal in building more resilient healthcare systems to better protect HCWs and patients, and to improve pandemic preparedness.’ (L555-561).

I agree about how this could be useful in some form of real time information system – but no real detail is presented as to how that could be achieved – even conceptually.

We have now edited the discussion to more specifically describe the concept. Specifically, a final product may include a dashboard, which we now explicitly refer to (L528-530 & 546-547).

So I’m left a little flat – I admire the topic, I do believe in its overall value, and I am impressed with the methods and application of the authors. But I’m not sure anyone reading the manuscript currently would be inspired to change operations.

We appreciate the reviewer’s thoughtful comments and hope our amendments have helped to alleviate these concerns. As we outline in the discussion, there are many challenges that need to be overcome before such a framework is operationalised, and this requires further research. However, we believe our framework provides the foundations for which the research community can build on to ultimately support routine operations and better protect staff and patients.

Reviewed again by Reviewer 3 who confirmed their comments had been addressed and recommended the manuscript for publication.